# Assessing the impact of Cyber-Physical Attacks on Critical Infrastructures: A Semantic-Based Approach

Mohamad Rihany, Fatma-Zohra Hannou, Nada Mimouni, and Fayçal Hamdi

CEDRIC lab, CNAM - Conservatoire National des Arts et Métiers Paris
mohamad.rihany@lecnam.net, fatma-zohra.hannou@lecnam.net,
nada.mimouni@cnam.fr, faycal.hamdi@cnam.fr

**Abstract.** This paper proposes an integrated approach to study impact propagation of cyber and physical incidents within critical healthcare infrastructures. This approach is based on a semantic modeling and reasoning engine which takes into account assets and input/output incident types while running propagation through a network graph. Besides, it calculates impact scores based on assets availability and protection degree upon incident reception.

## 1 Introduction

Over the last decade, many companies and organizations around the world have faced numerous threats that quickly increased in their magnitude and sophistication. The sources of these threats are heterogeneous. Indeed, as almost everything is today connected to the internet, an increasing risk of cyber-attacks are to be considered. But not only since physical attacks and intrusions are to be taken into account. For example, a fire can serve as a diversion for massive cyber attacks and theft of medical equipment. Thus treats cannot be analyzed solely as cyber or physical, and it is, therefore, essential to develop an integrated approach to fight against a combination of threats.

In the context of the EU Horizon 2020 SAFECARE project[1], we propose a solution to better understand the tight relationships between the assets' characteristics of a hospital's infrastructure and the propagation of attacks' effects to better prevent the impacts and consequences of incidents. Since these infrastructures host a variety of medical and IT assets with very different characteristics, an effective reaction to attacks needs to capture the detailed knowledge of intrinsic and contextual assets properties. Thus, we propose a model that is able to capture the essential characteristics related to incidents understanding and propagation and that takes into account the possible evolution of this knowledge. The impact propagation mechanism that we conceive considers the assets, their vulnerabilities, their interdependencies, their contextual knowledge, and the incidents that occurred in their environment.

---

[1] https://www.safecare-project.eu

The rest of this paper is organized as follows: In section 2, we present related work. In section 3, we describe the three components of our approach. Section 4 presents a use case. Finally, Section 5 draws conclusions and future research directions.

## 2   Related Work

There is an extensive work on the semantic modeling of cyber and physical security[2],and specifically, some domain ontologies for the security of healthcare systems [3]. They are either generic (core ontologies) or task-oriented covering risk analysis, information systems assessment, or prevention-oriented. In the same vein, the incident propagation study has been investigated in [1] and [5], to cite only few. Aside from semantic approaches, the work of [4] stands out among the first to integrate cyber-physical interdependencies for the estimation of the cascading effects of threats. In general, shoddy research work has been dedicated to combining semantic modeling of physical and cyber security to identify the physical, cyber interactions, allowing the expression of rules for propagating incidents and the generation of incidents impacts considering hybrid threats.

## 3   Semantic-based Impact Propagation Approach

The approach we proposed is three folds: (i) semantic modeling of the critical infrastructure (assets and their relationships), (ii) capturing of the expert's knowledge by generating generic rules that describe the propagation of cyber-physical attacks, (iii) assessing the impact a threat could have on different assets. The following sections detailed the different components of our approach.

### 3.1   Semantic Modeling

A modular ontology is designed based on the knowledge acquisition phase outcome, including three sub-ontologies: asset, impact, and protection. For lack of space, we gave a brief description of the asset sub-ontology. The following definitions have been mainly formulated by referring to existing security ontologies and risk management standards.

– $Asset \sqsubseteq \top$: an asset designates any valuable resource for an organization.
– $AccessPoint \sqsubseteq \top$: the access points are generally the gateways that enable the use of the resource, and by the same, the occurrence of the incident
– $Controller \sqsubseteq \top$: controllers are physical equipment or virtual protocols responsible implementing the restriction of access to assets, expressed on predefined access policies.
– $Device \sqsubseteq Asset$: refers to any tangible equipment, whether associated to a computer software with an automatic action (camera, sensor) or not (door).
– $Staff \sqsubseteq Asset$: represents any physical person performing regular or occasional tasks within the institution (hospital).

### 3.2   Impacts Propagation

To identify how impacts could propagate between different assets within a health infrastructure, we propose to extract from the different scenarios that reflect expert knowledge, a set of generic rules. These rules specify in which conditions incidents could propagate and impact different assets. The idea is to study all cyber and physical incidents and assets that belong to the same or close categories and identify the relationships that may convey the impacts. For example, in the scenario of a physical incident like fire, all physical and cyber assets that are related through a "`hosts`" relationship could be impacted. The role of the domain expert is therefore primordial to deal with this complexity.

### 3.3   Estimating Impact Score

The objective here is to assess the impact a threat could have on different assets within the end users' systems. Indeed, an asset could have different possible threats depending on the considered scenario and kill chain. For each threat, different possible protections are deployed by end users' systems. A value of protection, we call "protection degree", is calculated by experts for each asset per threat per protection. We define the impact score of an asset for a given threat and protection as follows:

$$impactScore_i(a) = 1 - protectionDegree_i(a) \tag{1}$$

Where, $i$ is a given threat, a is an asset. The impact score takes values between 0 and 1. Another case may occur when multiple protections are defined for one threat. In this case, the impact score is calculated as follows:

$$impactScore_i(a) = 1 - \sum_{j=1}^{p} protectionDegree_i(a) \tag{2}$$

Where, $p$ is the number of protections for an asset per threat $i$. When multiple protections are deployed on an asset, it becomes more robust against a possible threat. The aggregation of respective protection degrees reflects this phenomenon. When an incident comes on a system, it is transformed into threats according to the type of the asset and the asset source of the incident.

## 4   Use Case

In this part, a use case is presented to express the importance of the model in health Infrastructure. When the model receives an incident, it will be triggered to generate the impact propagation message containing the impacted assets. This is done by using the knowledge and the propagation rules that have been created before. We have worked on different heterogeneous scenarios between physical and cyber threats. In this paper, we present a simple scenario of physical threat only where a fire in a room will propagate to all the assets in this room.

The visualization of the propagation (cf. Figure 1) is made in a graphical way where the nodes represent the assets, and the edges represent the relationship between the assets. The impacted assets are expressed by changing the color of the node that represents this asset. The yellow represents the initial incident, red represents a strong impact, and orange represents a moderate impact. Assume that there is a fire in the asset "building1" as shown in Figure 1a: the color of this node will be changed into yellow as shown in the graph of Figure 1b. This incident will propagate in the room to affect the two assets "BMS_computer" and "Maintainer_Computer" which are colored orange and red, respectively.

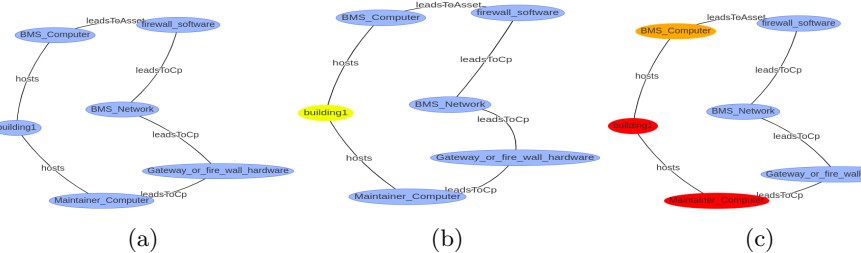

Fig. 1: Propagation of the Fire threat in the Network

## 5   Conclusion

This paper presents a semantic-based approach that assesses the impact propagation of complex cyber-physical attacks against critical infrastructure. This approach is currently being tested in a French hospital and will soon be evaluated on a larger scale (several hospitals within the European project).

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
