# OpenReview forum: "Assessing the impact of Cyber-Physical Attacks on Critical Infrastructures: A Semantic-Based Approach"
_eswc-conferences.org/ESWC/2021/Conference/Poster_and_Demo_Track — Submitted to ESWC2021 P&D_

### Official Review · AnonReviewer1 · 2021-04-14
**Interesting topic, novelty unclear**

**Rating:** 3
**Confidence:** 4

**Review:**

The paper addresses attacks in critical infrastructures. It acknowledges that physical and cyber attacks might be related and, based on that, builds a propagation network, based on semantic labels. It also defines impact scores to rate the potential gravity of an attack at the different parts of the system.

The authors mentioned in related work that this topic has been addressed by many in the past, but they don't discuss what are drawbacks of previous approaches and how their work can overcome them. Having this information is crucial to assess the novelty and thus the contribution of the work presented.

As a side note, the authors refer to previous work as "shoddy research". I personally think that criticism to fellow approaches should be done in a constructive way. Saying they were badly done, and not even justifying why, is not really appropriated.

In addition, Equation 2 seems to have a mistake, as I don't see the variable "p" in the sum function.


**Anonymity:**

Yes, I would like my review to remain anonymous.

---

### Official Review · AnonReviewer3 · 2021-04-14
**Quite vague presentation and unclear maturity of the proposed solution**

**Rating:** 5
**Confidence:** 4

**Review:**

The paper describes an approach to assess the impact of cyber-physical attacks on critical infrastructures.

Since the paper does not describe a demo explicitly (see also submission guidelines), I have understood that the paper presents a poster and not a demo

PROS

The topic is interesting and describes an interesting application scenario for semantic technologies (although not particularly new in terms of the type of supported tasks).

The paper describes the requirements and the goal of the proposed solution

CONS

My main concern is about the description of the approach, which is quite vague and does not adequately clarify the maturity of the proposed solution. Is the solution in the design phase or is the use case implemented and now under testing (like some statements lead to think)? Even in a poster, an even initial implementation is important as a (preliminary) testbed for the proposed idea.  This uncertainty in assessing the maturity of the proposed solution is also reflected in my confidence score.

If an implementation exists, I had expected a more quantitative description of the current use case: how many nodes are currently stored in the knowledge base? How many classes? How many triples? How many rules?

A link to the actual ontology is missing.

Details about the rules are missing. For example, which language is used for representing the rules?

It is not clear to me how impact propagation and score estimation are combined.

OTHER COMMENTS

In the related work, I suggest referring also to ontologies built based on STIX 2 - a reference model in use for cyber security data. For example, check UCO: unified cyber security ontology. (Syed et al. 2016)

“responsible implementing“ → responsible  for implementing

“A value of protection, we call “protection degree”, is calculated by experts” → It is not clear how experts can assess this degree. Also, it is not clear which kinds of experts are in charge of defining this... IT staff? Medical staff? Please explain at least the principles used for this assessment, otherwise, the explanation is too vague.


**Anonymity:**

Yes, I would like my review to remain anonymous.

---

### Official Review · AnonReviewer4 · 2021-04-15
**The article presents an approach to manage the knowledge related to assets, incidents and threads in the context of healthcare systems. The article is well-written and the application domains is relevant. However, there is not clear justification about the use of semantic web technologies.**

**Rating:** 5
**Confidence:** 4

**Review:**

Quality: Fair

Clarity: Low

Originality: Fair

Significance: Low

Pros
- The article is well-written.
- The application domain (healthcare systems) is relevant.
- The research is supported by a EU project.

Cons
- The use of semantic (web) technologies is not clear nor discussed.
- Lack of detail in the descriptions (e.g. representation and use of rules for incident propagation).
- The related work section does not specify the problems of the current approaches.

**Anonymity:**

Yes, I would like my review to remain anonymous.

---

### Official Review · AnonReviewer2 · 2021-04-15
**review of poster Assessing the impact of Cyber-Physical Attacks on Critical Infrastructures: A Semantic-Based Approach**

**Rating:** 5
**Confidence:** 5

**Review:**

In this poster the authors describe a semantic model for cybersecurity threats to a health care system. Such model is an ontology with rules. The authors also define a metric to represent how protected is a system. However the authors do not specify the details of that metric (which depend from the "experts"). Overall I think this is not novel nor documented enough.

**Anonymity:**

Yes, I would like my review to remain anonymous.

---

### Decision · Program_Chairs · 2021-04-19

Reject